# Early Microbial–Immune Interactions and Innate Immune Training of the Respiratory System during Health and Disease

**DOI:** 10.3390/children8050413

**Published:** 2021-05-19

**Authors:** Gustavo Nino, Carlos E. Rodriguez-Martinez, Maria J. Gutierrez

**Affiliations:** 1Division of Pediatric Pulmonary and Sleep Medicine, Children’s National Hospital, George Washington University, Washington, DC 20052, USA; 2Department of Pediatrics, School of Medicine, Universidad Nacional de Colombia, Bogota 111321, Colombia; carerodriguezmar@unal.edu.co; 3Department of Pediatric Pulmonology, School of Medicine, Universidad El Bosque, Bogota 110121, Colombia; 4Division of Pediatric Allergy and Immunology, Johns Hopkins University, Baltimore, MD 21218, USA; mgutie10@jhmi.edu

**Keywords:** microbiome, early-life, bronchiolitis, asthma

## Abstract

Over the past two decades, several studies have positioned early-life microbial exposure as a key factor for protection or susceptibility to respiratory diseases. Birth cohorts have identified a strong link between neonatal bacterial colonization of the nasal airway and gut with the risk for respiratory infections and childhood asthma. Translational studies have provided companion mechanistic insights on how viral and bacterial exposures in early life affect immune development at the respiratory mucosal barrier. In this review, we summarize and discuss our current understanding of how early microbial–immune interactions occur during infancy, with a particular focus on the emergent paradigm of “innate immune training”. Future human-based studies including newborns and infants are needed to inform the timing and key pathways implicated in the development, maturation, and innate training of the airway immune response, and how early microbiota and virus exposures modulate these processes in the respiratory system during health and disease.

## 1. Introduction

The past 30 years have revolutionized our view of the early development and maturation of the immune system in humans. Early studies introduced the concept of the “hygiene hypothesis”, following the observation that extremely clean household environments during infancy may increase the risk of atopy, asthma, and other respiratory conditions [1,2,3]. More recently, translational studies have confirmed that microbial exposure taking place during early life is essential in order to shape innate immune responses and modulate the risk of childhood asthma [4]. The contemporary view of these findings is conceptualized under the term innate immune training (IIT) [5,6,7,8], which refers to the environmental programming of immune responses of the epithelium and innate immune cells [5]. In contrast to adaptive immune responses, epigenetic reprogramming—rather than gene recombination—is believed to be the primary molecular mechanism of IIT [5]. Thus, exposure to bacteria and other environmental factors during infancy may play a fundamental role in modulating long-term immune responses to subsequent exposures [9,10]. IIT may explain, at least in part, the strong link between neonatal bacterial colonization of the airways and gut, and individual susceptibly to respiratory infections and childhood asthma. IIT may also be implicated in the complex interaction between the airway microbiome and viral infections during early life. The goal of this article is to review the existing literature on early microbial–immune interactions in the context of the new IIT paradigm during the newborn period and the first 2–3 years of life (Figure 1), which offers promising opportunities to understand and prevent respiratory disorders during infancy and beyond childhood.

## 2. Neonatal Nasal Airway Microbial Populations and Respiratory Outcomes Later in Life

Several studies have demonstrated that early-life microbial environments are critically associated with developing protection or susceptibility to respiratory diseases (Figure 2). Some of the early evidence demonstrating this important notion includes the seminal work from the Copenhagen Prospective Study on Asthma in Childhood birth cohort (COPSAC) in Denmark [11]. Bisgaard et al. made the critical observation that the bacterial colonization of the hypopharynx with pathogenic bacteria during the newborn period was related to subsequent clinical outcomes [11]. Examining samples from 321 neonates at one month of age, they identified that infants with nasal colonization with S. pneumoniae, H. influenzae, or M. catarrhails, or with a combination of these organisms, but not S. aureus, are at increased risk for recurrent wheeze and asthma by 5 years of age [11]. A subsequent prospective study of 411 infants with a maternal history of asthma identified that early airway colonization with S. pneumoniae, H. influenzae, or M. catarrhalis was linked to a higher risk of pneumonia and bronchiolitis during infancy [12]. The notion that neonatal airway microbial populations predict respiratory outcomes later in life has been confirmed by additional birth cohorts and other longitudinal studies. A Moraxella-dominant nasal microbiota profile in early childhood was linked to an increased rate of respiratory infections within the first two years of life, according to a population-based birth cohort analysis in Finland (*n* = 839) [13]. Follow-up studies of this cohort demonstrated that a persistent Moraxella profile was associated with a greater asthma risk, and similar results were observed between the longitudinal variations in nasal microbiota during age 2 to 24 months old and asthma risk later in life [14]. The Australian Childhood Asthma Study (CAS; *n* = 234 children) and the Growing Up in Singapore Towards Healthy Outcomes (GUSTO) birth cohort (*n* = 122) have reported similar results [15,16]. Collectively, these large longitudinal studies provide strong support to the hypothesis that the neonatal nasal microbiome is critically involved in the development of protection or susceptibility to infant respiratory infections, early-life wheezing, and childhood asthma later in life.

## 3. Nasal Airway Microbiota–Immune Interactions in Early Infancy

Although the mechanism underlying the link between early pathogenic airway bacterial colonization and subsequent susceptibly to respiratory infections and asthma is unknown, current evidence suggests that a pro-inflammatory immune signature of the respiratory mucosa is implicated. A study examining cytokines and chemokines in the neonatal nasal airway (*n* = 662) showed that colonization with M. catarrhalis and H. influenzae induced a mixed Th1/Th2/Th17 response with high levels of IL-1β, TNF-α, and macrophage inflammatory protein-1β [17]. Interestingly, while S. aureus colonization demonstrated a Th17-promoting profile with elevated IL-17 levels, S. pneumoniae colonization was not significantly associated with any of the mediators [17], suggesting a distinct mechanism for the link between S. pneumoniae and respiratory outcomes. Another study including nasal microbiome analyses of 700 children monitored since birth showed that a higher relative abundance of Veillonella and Prevotella in the airways at one month old is associated with increased asthma risk [18]. This asthma-associated composition is correlated with lower levels of IL-1β and TNF-α, which may characterize an inefficient anti-bacterial response, but significantly higher levels of monocyte and T-cell recruiting chemoattractants (CCL2 and CCL17) [18]. These studies indicate that the nasal bacterial colonization of neonates is associated with a distinct immune signature of the airway mucosa that may alter subsequent responses to environmental challenges.

There is also recent evidence suggesting that the nasal airway of human infants has developmental features that makes it intrinsically pro-inflammatory. Our team has shown that compared with older children, human infants (<18 months) exhibit increased production of type III IFN-lambda during viral respiratory infections [19]. Type III IFN-lambda is an antiviral molecule produced almost exclusively by the epithelium [20,21] and it has an intriguing role maintaining the bacterial microbiome [22,23]. Our in vivo findings were complemented by an in vitro demonstration of strong type III IFN-lambda production in human infant nasal epithelial cells via the pro-inflammatory activation of NF-kB signaling induced by either IL-1β or TLR-3 agonist exposure [19]. We also observed that human infant nasal airway epithelial cells exhibit a strong NF-kB-driven production of the master type 2 cytokine thymic stromal lymphopeitin (TSLP), which was linked to increased severe respiratory infections during the first two years of life [24]. TSLP is produced in response to allergens and pathogens such as RSV [25] and rhinovirus [26,27], and it has been implicated in type 2 diseases, such as allergic asthma [25]. Thus, the interaction of viral and bacterial pathogens with strong NF-kB-mediated responses in the infant airway epithelium may lead to a pro-inflammatory state in the respiratory mucosa.

Aside from nasal mucosal responses, the in vitro systemic immune response to Haemophilus influenzae, Moraxella catarrhalis, and Streptococcus pneumoniae in the peripheral blood mononuclear cell of infants is also predictive of lower respiratory infection [28]. The increase of plasma inflammatory biomarkers in early life, such as C-reactive protein, IL-1β, IL-6, TNF-α, and CXCL8 (IL-8), has also been linked to reduced neonatal lung function [29]. More than a single cytokine, a set of distinct immune phenotypes are seen in infants developing asthma during childhood. While neutrophil-linked antiviral responses provide an increased risk of transient asthma, persistent asthma at 6 years old is characterized by early life bacterial colonization and type 2 responses (enhanced IL-5 and IL-13 production) in stimulated T cells at 18 months of age [30]. These findings suggest a mechanism of microbiota–immune interactions in early infancy driven by intrinsic developmental differences in the nasal mucosa and systemic immune responses modulated by early bacterial environments. An overabundance of pathogenic microbiota coupled with an airway pro-inflammatory state in infancy may predispose some children to respiratory infections, wheezing illnesses, and asthma later in life.

## 4. Neonatal Intestinal Microbial Populations and Respiratory Outcomes Later in Life

Because of its location, the nasal microbiome is likely to play a direct role in the development of immune responses in the respiratory tract. Nonetheless, the gut microbiota is much more diverse and complex, and there is strong evidence that is linked to distinct immunological responses during health and disease [31]. The intestinal microbiome is also under the influence of many environmental factors beginning in early life, including diet and medications, particularly antibiotics [31]. These early environmental factors may alter the natural history of the infant gut microbiome development [31], with potential consequences for the immune responses and the individual predisposition to respiratory disease. Indeed, the importance of a bidirectional crosstalk, conceptualized as the gut–lung axis, has been increasingly recognized as a critical contributor to respiratory health [32].

Some of the most compelling evidence or the gut–lung axis, driven by changes in the intestinal microbiota, comes from the Canadian Healthy Infant Longitudinal Development (CHILD) cohort. The CHILD cohort (*n* = 319) found that transient gut microbial dysbiosis in the first 100 days of life is linked to asthma risk. The CHILD study also demonstrated that Faecalibacterium, Lachnospira, Veillonella, and Rothia are protective against asthma [33]. The investigators confirmed this association in an animal model, in which the administration of these protective bacterial taxa reduced lung inflammation in previously germ-free mice inoculated with stool from an asthmatic patient [33]. The link between early abnormalities of the gut–lung axis and subsequent asthma risk is also supported by a large prospective birth cohort of 700 children, in which specific gut microbiome signatures at 1 year old were associated with asthma at age 5 [34]. Translational longitudinal studies have allowed for a better understanding of the maturation of the gut microbiome and the mechanistic link with subsequent respiratory outcomes. The longitudinal analyses of 100 newborn children sampled multiple times during their first 3 months of life, reported by Oli et al. [35], identified that the stereotypic pattern of immune development appears to be driven by microbial interactions and hampered by early gut bacterial dysbiosis [35]. This seminal study showed that exceptionally low gut microbiome diversity (near-complete dominance of the gut microbiome by bacterial classes Bacilli or Gammaproteobacteria) was associated with more circulating activated T cell populations in their subsequent three-month blood samples [36]. Another seminal study, the Protection against Allergy: Study in Rural Environments (PASTURE) birth cohort, modeled the early maturation of the intestinal microbiome during the first year of life. In 12-month-old infants, the approximate “microbiome age” was related to previous farm exposure and a lower risk of childhood asthma. Interestingly, PASTURE investigators identified inverse associations of asthma with fecal butyrate, the bacterial taxa that predict butyrate production and the expression of butyryl-coenzyme A (CoA): acetate-CoA-transferase, a major enzyme in butyrate metabolism [36]. A different study identified that a fecal metabolite produced by intestinal bacteria (12,13-diHOME) has potent immunogenic properties. as it inhibits the number Treg cells and promotes airway inflammation in vitro and in vivo in an animal model [37]. Complementary analyses of two birth cohorts found that the increased concentration of fecal 12,13-diHOME during the neonatal period is linked to an increased risk of atopy, eczema, or asthma [37]. Similarly, another study examining the mechanistic link between the gut microbiome and metabolome identified that microbiota-derived acetate protects against respiratory syncytial virus infection through a GPR43-type 1 interferon response [38]. Taken together, these studies support the hypothesis that the gut microbiome contributes to respiratory health protection, shaping early immune development via microbiome-related metabolites.

## 5. The Symbiosis of Bacterial Microbiota and Respiratory Viruses in Early-Life

The traditional view that respiratory viruses are primary triggers of wheezing while bacteria are mostly linked to pneumonia is increasingly being re-evaluated. Several studies have shown the coexistence of virus and bacteria in the airways with important clinical and pathobiological implications [39]. A study including asthmatic children (*n* = 361, 984 samples) identified that wheezing episodes were associated with both bacterial infection (odds ratio 2.9, *p* < 0.001) and virus infection (odds ratio 2.8, *p* < 0.001) [40]. In another study, Teo et al. examined the longitudinal dynamics of the nasal microbiome profiles in young infants (*n* = 234) and identified that temporary overabundance of Streptococcus, Moraxella, or Haemophilus occurred during viral respiratory infections [15]. Mansbach et al., in a multi-centric longitudinal cohort (*n* = 842 infants hospitalized for bronchiolitis), established that the nasal microbiome enrichment of Moraxella or Streptococcus species after bronchiolitis hospitalization was associated with recurrent wheezing by the age of three [41]. Zhang et al. also found that in infants hospitalized during the initial episode of severe RSV bronchiolitis at six months of age or less (*n* = 74), the relative abundance of Haemophilus, Moraxella, and Klebsiella was linked to recurrent wheezing [42]. Another longitudinal study by Rosas-Salazar (*n* = 118) identified that the during RSV acute respiratory infection in infancy, nasopharyngeal identification and increased abundance of Lactobacillus were linked to a lower risk of childhood wheezing illnesses at age 2 [43]. Thus, extensive evidence exists of a virus–bacteria interaction that is strongly associated with the outcome of childhood respiratory infections.

Mechanistic insights into the relationship between viruses and bacteria during respiratory infections in children have been examined using transcriptomic profiles and companion clinical outcomes. A translational study by de Steenhuijsen Piters [44] identified that in young children with RSV infection, hospitalization was linked to H. influenzae and Streptococcus, and was negatively associated with S. aureus. Additional analyses showed that the transcriptome profiles of children with RSV infection, as well as H. influenzae- and Streptococcus-dominated microbiota, were characterized by the overexpression of genes linked to TLR and neutrophil activation [44]. A recent mechanistic study also demonstrated that the molecular markers of neutrophilic inflammation in the respiratory mucosa enhance RSV clinical severity [45], which is line with prior reports linking severe RSV infection to neutrophilic inflammation [46,47]. These translational studies demonstrate that the symbiotic relationship of bacteria and viruses in early-life determines the type of individual airway immune responses during acute respiratory infections, and may have critical implications for acute and longitudinal clinical outcomes during infancy and beyond childhood.

Although most studies have focused on how bacteria may influence the outcomes of viral infections, it is important to mention that there is also evidence of a bidirectional interaction. A multicenter, single-blind, randomized, placebo-controlled trial to evaluate RSV palivizumab prophylaxis (*n* = 429) demonstrated that a virus-targeted intervention shaped the airway microbiome [48]. A comparison with placebo at 1 year old demonstrated that the palivizumab-treated infants showed a significantly lower abundance of Staphylococcus, an increased abundance of other species (e.g., Klebsiella), and an increased diversity of oral taxa [48]. At 6 years old, the palivizumab group showed a greater abundance of Haemophilus and a lower abundance of Moraxella and Neisseriaceae compared with the placebo-treated children [48]. These results demonstrate that viruses and bacteria not only coexist during respiratory infections, but have bidirectional interactions. Modulation of these virus–bacteria interactions may be critically important to improve the clinical outcome of infants and asthmatic children, as demonstrated by the positive results of recent randomized, placebo-controlled trials aimed to assess the beneficial effect of azithromycin during viral respiratory illnesses in young children [49,50,51,52]. Notably, recent findings suggest that azithromycin likely operates through antimicrobial actions, as the airway microbiota composition in acute episodes of asthma-like symptoms modifies the effect of azithromycin treatment [53,54].

## 6. Microbiome-Driven Interventions and Early Respiratory Health: The Novel Notion of Innate Immune Training

All of the evidence from birth cohorts and companion mechanistic studies indicate that the early microbiome shapes immune system development, and may increase or reduce susceptibility to respiratory diseases, including childhood asthma. This new paradigm has been conceptualized under the emergent model of “Innate Immune Training (IIT)”, in which microbiota and epigenetic influences are convergent drivers and mediators of the asthma trajectory from pregnancy to childhood [6,7,8,9,10]. In the same way that pathogenic bacteria during early life may increase the risk for respiratory disorders, exposure to protective bacteria may serve as a primary prevention. A proof of principle for this model has been recently provided by Nieto et al. in a randomized placebo-controlled clinical trial (*n* = 120) [55]. In this study, mucosal sublingual immunotherapy based on whole inactivated bacteria (MV130) reduced the number of wheezing attacks (WA) and other secondary outcomes [55]. While this study supports the notion of IIT, it is still unclear regarding what the best timing for intervention should be. Recent evidence suggests that the nasal microbiome is affected by environmental exposure in utero. Hjelmsø et al. analyzed the microbiome data specimens from 695 pregnant women and their offspring obtained during a randomized trial of n-3 long-chain fatty acids and vitamin D. The results identified that these maternal dietary interventions during pregnancy modified the infant airways (microbiome and immune profile), but not the infant fecal or maternal vaginal microbiota [56]. As cesarean section changes the infant’s gut microbiome and increases the risk of asthma [57], novel approaches are being developed to modeling the transfer of vaginal microbiota from mother to infant in early life as a potential new therapeutic strategy [58].

## 7. Conclusions

Several human-based translational studies have provided strong support to the concept that microbial and other environmental exposures taking place during early life, even in utero, can affect the biology of the mucosal barriers and drive the development and maturation of immune responses in the respiratory system. Future human-based studies including newborns and infants are still needed to inform the timing and key pathways implicated in the development, maturation and innate training of the airway immune response, as well as how early microbiota and viruses exposure modulates these processes in the respiratory system during health and disease.

## Figures and Tables

**Figure 1 children-08-00413-f001:**
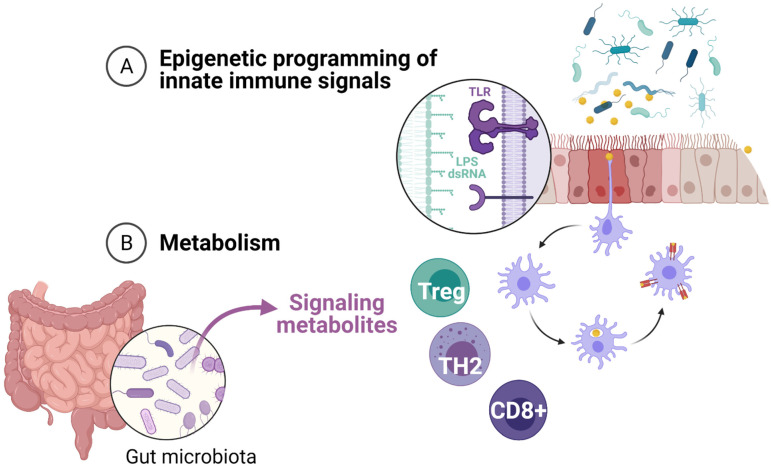
Early innate immune training of the respiratory system. Mechanisms of innate immune training include (**A**) direct environmental programming of the airway epithelium and innate immune cells in the respiratory mucosa, and (**B**) systemic effects driven by gut microbiota signaling metabolites.

**Figure 2 children-08-00413-f002:**
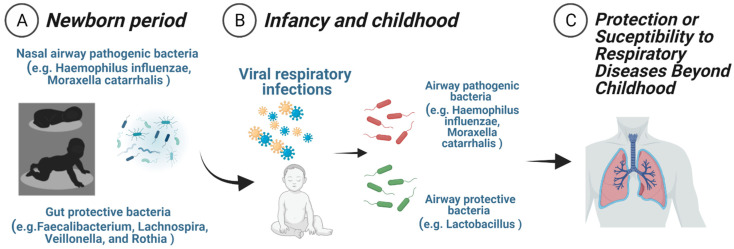
Early microbial–immune interactions modulate risk against respiratory diseases. Early microbial–immune interactions include (**A**) the newborn period (3 months), during which initial bacterial colonization of the nasal airways and gut are essential to shape respiratory immunity, and (**B**) infancy and early childhood (first 2–3 years of life), where transient microbiome changes during viral respiratory infections are linked to distinct airway immune responses and respiratory outcomes. (**C**) The events occurring during these periods of innate immune training play a crucial role in the development of asthma and other respiratory conditions beyond childhood.

## Data Availability

None of this is applicable.

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
