# Peer review of "Early Microbial–Immune Interactions and Innate Immune Training of the Respiratory System during Health and Disease"

_children, 2021, doi:10.3390/children8050413_

Round 1
Reviewer 1 Report
- The abstract need to be re-write to avoid similarity with the introduction.
- In the abstract, the authors mentioned that they will focus on “Innate immune training”. The authors should define the IIT in the introduction including their role in the lower and upper airway infection.
- The paper focus on the number of study participants in all listed Trails, the outcomes or the frequencies (infected or diseased compared to controls) give a sound argument for the paper.
- Nasa airway microbiota-immune interaction and bidirectional interaction of viral-bacterial need to be summarized in one figure or separate figures.
- Lines (49 and 51) S. pneumoniae was a risk of recurrent wheeze and asthma by 5 years of age. In lines (77-78), S. pneumoniae had no not significantly associated with any of the mediators. Could you explain the difference in these two findings?
Author Response
Response: We thank the reviewer for the helpful and thoughtful comments made regarding our manuscript.
Comments and Suggestions for Authors
- The abstract need to be re-write to avoid similarity with the introduction.
We have re-written the introduction to include definition of IIT and avoid similarities with the abstract.
- In the abstract, the authors mentioned that they will focus on “Innate immune training”. The authors should define the IIT in the introduction including their role in the lower and upper airway infection.
We have addressed this issue as stated above.
- The paper focus on the number of study participants in all listed Trails, the outcomes or the frequencies (infected or diseased compared to controls) give a sound argument for the paper.
We have modified the introduction to state the goal of the paper and provide better rationale for the link of early microbial immune interactions and IIT.
- Nasa airway microbiota-immune interaction and bidirectional interaction of viral-bacterial need to be summarized in one figure or separate figures.
We added Figure 1 to provide an overview of the concept presented in the paper.
- Lines (49 and 51) S. pneumoniae was a risk of recurrent wheeze and asthma by 5 years of age. In lines (77-78), S. pneumoniae had no not significantly associated with any of the mediators. Could you explain the difference in these two findings?
We added a line to explain that these findings suggest “a distinct mechanism for the link between S. pneumoniae and respiratory outcomes.”
Thank you.
Reviewer 2 Report
There are some key references that could be included:
Origins of the hypothesis. Acknowledge:
- DP. Strachan. Hay fever, hygiene, and household size. BMJ. 1989 Nov 18; 299(6710): 1259–1260
- von Mutius E, Fritzsch C, Weiland SK, Röll G, Magnussen H.Prevalence of asthma and allergic disorders among children in united Germany: a descriptive comparison. BMJ. 1992 Dec 5;305(6866):1395-9. doi:10.1136/bmj.305.6866.1395.
Review data from farming communities. Consider:
1. Stein MM, Hrusch CL, Gozdz J, Igartua C, et al. Innate Immunity and Asthma Risk in Amish and Hutterite Farm Children.
N Engl J Med. 2016 Aug 4;375(5):411-421.
Consider the role of the microbial metabalome an increasingly important field.
1.Levan SR, Stamnes KA, Lin DL Panzer AR. Elevated faecal 12,13-diHOME concentration in neonates at high risk for asthma is produced by gut bacteria and impedes immune tolerance. Nature Microbiology 2019; 4, 1851–61.
Methodology for measuring the microbiome could be a separate section.
Author Response
Comments and Suggestions for Authors
Response: We thank the reviewer for the helpful and thoughtful comments made regarding our manuscript.
There are some key references that could be included:
Origins of the hypothesis. Acknowledge:
- DP. Strachan. Hay fever, hygiene, and household size. BMJ. 1989 Nov 18; 299(6710): 1259–1260
- von Mutius E, Fritzsch C, Weiland SK, Röll G, Magnussen H.Prevalence of asthma and allergic disorders among children in united Germany: a descriptive comparison. BMJ. 1992 Dec 5;305(6866):1395-9. doi:10.1136/bmj.305.6866.1395.
Review data from farming communities. Consider:
- Stein MM, Hrusch CL, Gozdz J, Igartua C, et al. Innate Immunity and Asthma Risk in Amish and Hutterite Farm Children.
N Engl J Med. 2016 Aug 4;375(5):411-421.
Consider the role of the microbial metabalome an increasingly important field.
1.Levan SR, Stamnes KA, Lin DL Panzer AR. Elevated faecal 12,13-diHOME concentration in neonates at high risk for asthma is produced by gut bacteria and impedes immune tolerance. Nature Microbiology 2019; 4, 1851–61.
Response: We have added all the references and included the concepts suggested by the reviewer in the introduction.
Methodology for measuring the microbiome could be a separate section.
Response: We feel that specific methodology of microbiome is beyond the scope of the review and prefer to maintain sections focused on the discussion of clinical implications of the early microbial-immune interactions in the context of the new IIT paradigm
Thank you.
Round 2
Reviewer 1 Report
In lines 19 and 20, the authors pointed that "In this review we summarize and discuss our current understanding of how early microbial-immune interactions occur during infancy, with a particular focus on the emergent paradigm of “innate immune training”. ".
Thank you for providing a figure about microbial-immune interactions. As the authors mentioned in 238, please add a separate figure for “Innate Immune Training (IIT)”.
Author Response
COMMENT: In lines 19 and 20, the authors pointed that "In this review we summarize and discuss our current understanding of how early microbial-immune interactions occur during infancy, with a particular focus on the emergent paradigm of “innate immune training”. ".
Thank you for providing a figure about microbial-immune interactions. As the authors mentioned in 238, please add a separate figure for “Innate Immune Training (IIT)”.
RESPONSE: We have modified the manuscript to include a separate figure describing mechanisms of innate immune training (Figure 1) in addition to early microbial-immune interactions (Figure 2). Thank you.